# NGS method for parallel processing of high quality, damaged or fragmented input material using target enrichment

**Anna Lyander**[1,2]⊛*, **Anna Gellerbring**[2]⊛, **Moa Hägglund**[2], **Keyvan Elhami**[2], **Valtteri Wirta**[1,2]

**1** Science for Life Laboratory, KTH Royal Institute of Technology, Clinical Genomics Stockholm, School of Engineering Sciences in Chemistry, Biotechnology and Health, Stockholm, Sweden, **2** Department of Microbiology, Science for Life Laboratory, Karolinska Institutet, Clinical Genomics Stockholm, Tumor and Cell Biology, Stockholm, Sweden

⊛ These authors contributed equally to this work.
* anna.lyander@scilifelab.se

**Data Availability Statement:** All relevant data are within the manuscript and its Supporting Information files.

## Abstract

Next-generation sequencing (NGS) has been increasingly popular in genomics studies over the last decade and is now commonly used in clinical applications for precision diagnostics. Many disease areas typically involve different kinds of sample specimens, sample qualities and quantities. The quality of the DNA can range from intact, high molecular weight molecules to degraded, damaged and very short molecules. The differences in quality and quantity pose challenges for downstream molecular analyses. To overcome the challenge with the need of different molecular methods for different types of samples, we have developed a joint procedure for preparing enriched DNA libraries from high molecular weight DNA and DNA from formalin-fixed, paraffin-embedded tissue, fresh frozen tissue material, as well as cell-free DNA.

## Introduction

Next generation sequencing (NGS) is widely used for clinical applications in precision diagnostics [1], including for example diagnostic applications for rare inherited diseases and cancer, as well as therapy selection or monitoring for cancer patients [2,3]. A key step in the NGS process is the conversion of the extracted nucleic acid into molecular libraries compatible with the sequencing platform. Several methods are available for this conversion step, referred to as library preparation, with key differences in several aspects.

Targeted sequencing is a particularly interesting NGS application since it enables analysis of a predefined part of the genome at a high sequencing depth, which is required for low allele frequency variant detection, while retaining an acceptable cost profile. Target enrichment by DNA hybridization (also known as hybrid capture) in combination with short-read sequencing is a technique that has proven to be suitable for clinical sequencing [1].

A common challenge for the different library preparation methods is, however, the dependency on input material. Depending on the source and pre-analytical processing of the sample,

**Funding:** The author(s) received no specific funding for this work.

**Competing interests:** Valtteri Wirta has received reimbursement of travel costs and speaker's honoraria from Illumina. This does not alter our adherence to PLOS ONE policies on sharing data and materials.

the input material can be intact, high molecular weight DNA (e.g., from blood or bone marrow), degraded and damaged from formalin-fixed, paraffin-embedded (FFPE) specimens [3], or fragmented in short molecules as from liquid biopsies in the form of cell-free DNA (cfDNA) [4]. The differences in quality and quantity pose challenges for downstream molecular analyses of the DNA molecules, and the protocols for library preparation methods have therefore to be adapted to accommodate these differences.

Ideally, for enabling a cost-efficient and rapid processing in a clinical diagnostic laboratory, the library preparation, enrichment and sequencing strategy chosen should be suitable for the vast majority of applications that can be envisioned, including compatible with blood, FFPE, fresh frozen (FF) tissue material as well as with cfDNA.

Here, we report a method for preparing enriched DNA libraries for targeted sequencing from high molecular weight DNA and DNA from blood, FFPE and FF material as well as cfDNA in parallel using the same protocol.

## Materials and methods

### Material

Four poor quality FFPE samples with DNA integrity number (DIN) 1.0 (denoted FFPE-1), 1.1 (FFPE-2), 1.0 (FFPE-3), 1.7 (FFPE-4), the reference sample NA24143 (Coriell Institute), and four commercially sourced reference samples HD798, HD799, HD803, HD777 (Horizon Discovery) were included in the study. Samples HD798, HD799, HD803 are Quantitative Multiplex Formalin Compromised DNA Reference Standards which contain eleven mutations at varying allele frequencies. The allelic frequency of variants in the control samples have been determined by digital droplet PCR (ddPCR) by the manufacturer. These formalin-compromised reference samples vary in levels of fragmentation and formalin damage; mild (HD798, DIN 7.3), moderate (HD799, DIN 3.9) and severe (HD803, DIN 1.7). Sample HD777 is a cfDNA reference standard which contains eight mutations at 5.00% allele frequency. The variants in HD777 have been confirmed by ddPCR by the manufacturer.

### Ethical approval

Ethical approval (2022-01088-01, *Kvalitetssäkring av genetiska analyser med fokus på helgenomsekvensering och andra breda analyser*) has been approved for the samples used in this study.

### Sample buffer and buffer exchange

Samples may be stored in a buffer containing EDTA or other agents inhibiting enzymatic activity to provide stability and to prevent degradation of the DNA. However, these chemicals intercalate with ions also needed for enzymatic activities in the library preparation process. In order to obtain a highly complex library, the DNA is recommended to be eluted in 10 mM Tris-HCl pH 8–8.5 or molecular biology grade water. If the DNA has been eluted in a non-compatible sample buffer, a buffer exchange is recommended by using 1.8X sample-to-bead ratio of AMPure XP beads (Beckman Coulter) and a prolonged binding time (15 min) followed by two washes with freshly made 80% ethanol. A bead drying time of 1–3 min for ethanol evaporation is recommended since overdrying may lead to difficulty in retrieving high molecular weight DNA. Elute with 10 mM Tris-HCl pH 8–8.5 or molecular biology grade water in 80% of the sample volume for efficient yield. Note that buffer exchange is associated with a significant loss of DNA, especially if using a low initial volume.

## Library preparation

The library preparation method is a combination of KAPA HyperPrep and HyperPlus (Roche), which enables both high molecular weight DNA, damaged DNA (e.g., FFPE material) and cfDNA samples to be processed in parallel using the same protocol (Fig 1A). For a combined preparation of both genomic DNA (gDNA) and cfDNA, the HyperPlus kit (includes enzymatic fragmentation) is used. However, if there are solely cfDNA samples, the HyperPrep kit (without enzymatic fragmentation) should be used instead. The protocol has an adapter and indexing design which includes unique molecular identifier (UMI) sequence barcodes, providing the possibility to carry out sensitive analyses based on reads collapsing using the UMI barcodes.

The first step is enzymatic fragmentation, which is performed for all samples except cfDNA samples, and the following end-repair/A-tailing step is a combination of KAPA HyperPrep and HyperPlus protocols (hereafter denoted combination protocol). All cfDNA samples need a longer end-repair step to yield blunt ends for A-tailing and subsequent TA-ligation, while enzymatically fragmented samples can be processed with a significantly shorter step since the enzymatic fragmentation process leaves most molecular ends blunt. Despite this, we include an extended end-repair step also for other sample types since we hypothesize that short, damaged DNA molecules, e.g., DNA from FFPE material, cannot efficiently be fragmented enzymatically, however still could be turned into library molecules by performing a longer end-repair step prior to the A-tailing step. This combination also prompts that cfDNA samples can be handled in the same workflow. An extended end-repair step is not anticipated to have any negative effect on the blunt end molecules already fragmented by the enzyme in the KAPA HyperPlus kit. The end-repair and A-tailing (ERAT) enzyme mix with a purple cap (KAPA HyperPlus kit, Roche v 5.19) needs to be used for this protocol since the ERAT enzyme mix with an orange cap in the KAPA HyperPlus kit, despite its name, does not have end-repair activity.

## Enzymatic fragmentation of gDNA

The fragmentation step is only used for gDNA samples (not cfDNA samples). An amount of 1–250 ng DNA can be used for library preparation with the protocol, Table 1. The input amount of DNA was 50 ng for the FFPE samples (FFPE-1, FFPE-2, FFPE-3, FFPE-4); 50 and 250 ng for HD798, HD799, HD803; 10, 50 and 250 ng for NA24143; and 10 and 30 ng for HD777. The DNA is diluted with water to 30 μl, then mixed with 1X KAPA Frag Buffer and 10 μl of KAPA Frag Enzyme in a total volume of 50 μl, and then incubated at 37°C for 12.5 or 17.5 min followed by an incubation at 70°C for 15 min to heat inactivate the fragmentation enzyme. The fragmentation time has been optimized to give an insert size of 250 bp for high molecular weight DNA and can be adjusted to specific needs.

## End-repair, A-tailing and ligation of all sample types

At this step, cfDNA samples are included in the protocol by diluting an amount of 1–250 ng (Table 1) with water to 50 μl. For end-repair and A-tailing, 7 μl ERAT buffer and 3 μl ERAT enzyme mix is added to each sample (also previously fragmented gDNA samples) and incubated at 20°C for 30 min (end-repair) followed by 65°C for 30 minutes (A-tailing). Next, xGen Duplex Seq adapters (3 bp long UMI's, Integrated DNA Technologies), 0.55 μM (for 25–250 ng input amounts) or 0.14 μM (for < 25 ng input amounts), (Integrated DNA Technologies) are used for the ligation that are supplemented by 30 μl ligation buffer and 10 μl of DNA ligase in a total volume of 110 μl. Ligation is performed at 20°C for 15 min. A bead cleanup step follows the ligation, and is done with 0.8X sample-to-bead ratio of AMPure XP beads (Beckman

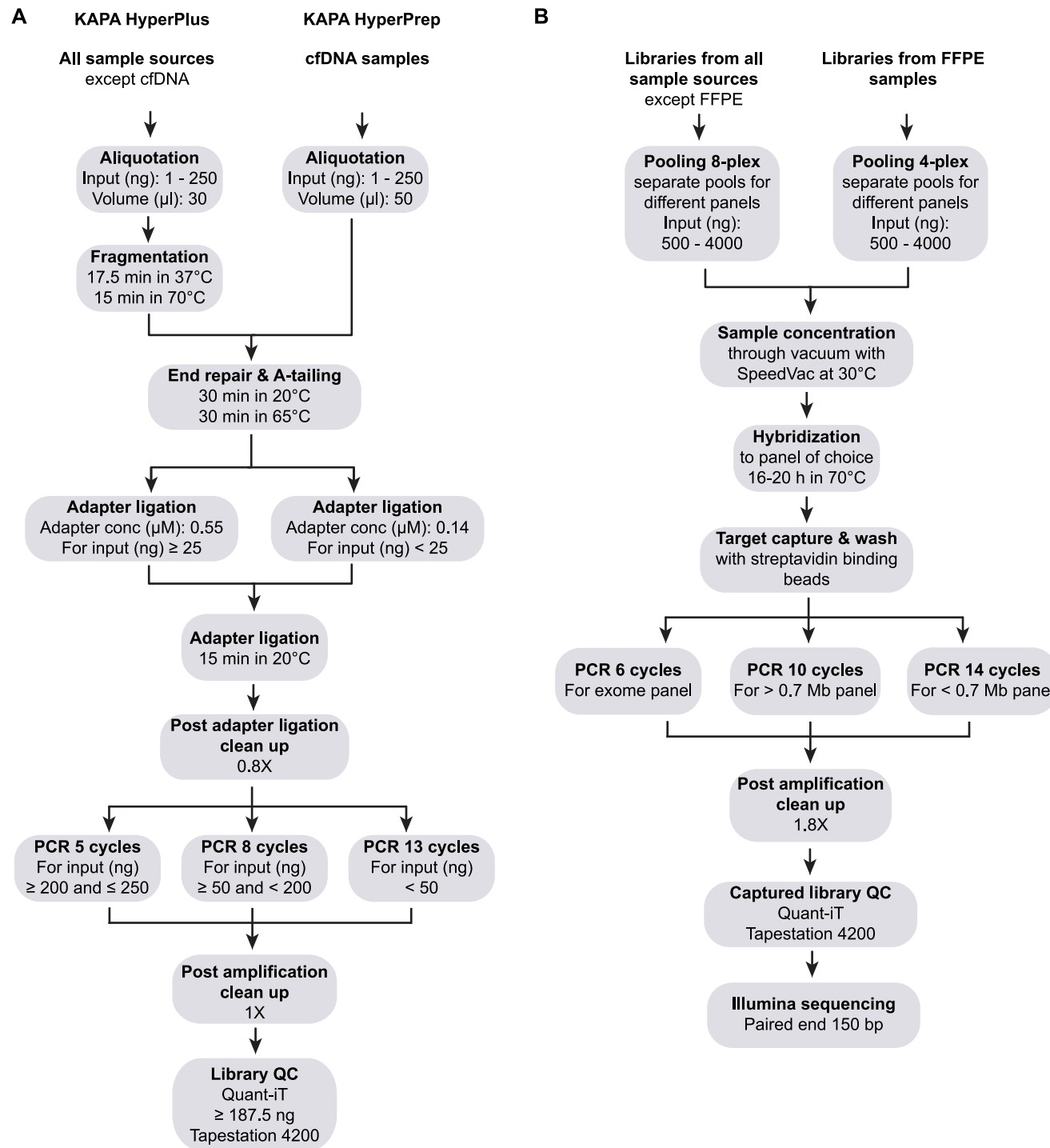

**Fig 1. Overview of method workflow.** Visualization of all steps for (A) library preparation and (B) target capture until sequencing. An alternative approach for the number of PCR cycles and input amount is: ≥ 150 ng and ≤ 250 ng, 5 cycles; ≥ 25 ng and < 150 ng, 8 cycles; < 25 ng, 10 cycles.

Coulter), binding at room temperature for 5 min, followed by two washes with freshly made 80% ethanol (VWR), drying 3–5 min and elution in 25 μl water, and final recovery of 20 μl supernatant.

**Table 1. Recommended input amount of DNA to library preparation.**

| Amount of DNA available (ng) | Input amount of DNA to prep (ng) |
|---|---|
| $\geq$ 250 | 250 |
| $<$ 250 | Use all available amount |

The recommendations of input amount of DNA to library preparation depends on the amount of DNA available of the sample.

## Indexing amplification

The fragmented (if needed), end-repaired, A-tailed and adaptor ligated molecules are subjected to a sample indexing amplification by adding 1X KAPA HiFi HotStart ReadyMix (Roche) and xGen indexing primers (2 μM, with 10-bp unique dual indices (UDIs), Integrated DNA Technologies) to 20 μl sample supernatant. PCR is run according to the following program: initial denaturation at 98˚C for 45 sec, X cycles (defined in Table 2) of: denaturation at 98˚C for 15 sec, annealing at 60˚C for 30 sec, elongation at 72˚C for 30 sec, and a final elongation at 72˚C for 1 min. The PCR amplification is followed by a bead clean up step with 1X sample-to-bead ratio AMPure XP beads (Beckman Coulter), binding at room temperature for 5 min, followed by two washes with freshly made 80% ethanol (VWR), drying 3–5 min and elution in 22 μl water, and final recovery of 20 μl supernatant. Prior to target enrichment the average library length is estimated by TapeStation assay D1000 (Agilent) and the library concentration is determined by Quant-iT dsDNA BR (ThermoFisher Scientific).

## Target enrichment

The target enrichment using hybrid-capture method is based on the protocol from Twist Bioscience and includes pooling, hybridization, target capture and wash (Fig 1B).

## Pooling and hybridization

Target enrichment can be made more cost-efficient by pooling several indexed libraries together and performing the target capture on several sample libraries simultaneously. A pool can contain up to eight indexed libraries in a total quantity of 500–4000 ng library. The recommended minimum amount required of an indexed library for pooling is 187.5 ng (1500 ng / 8 libraries = 187.5 ng per sample). For single-plex captures it is recommended to use 500 ng in total for balancing the ratio between indexed library, blockers and the capture probes. The amount of a library to a pool is decided from the amount of reads desired and the number of samples in the pool. For FFPE samples, it is recommended to only multiplex up to four indexed sample libraries due to the inherent quality differences between the samples and the

**Table 2. Recommendation of number of PCR cycles depending on input amount.**

| Alternative 1 | | Alternative 2 | |
|---|---|---|---|
| gDNA input amount (ng) | Number of PCR cycles | gDNA input amount (ng) | Number of PCR cycles |
| $\geq$ 200 ng and $\leq$ 250 ng | 5 | $\geq$ 150 ng and $\leq$ 250 ng | 5 |
| $\geq$ 50 ng and $<$ 200 ng | 8 | $\geq$ 25 ng and $<$ 150 ng | 8 |
| $<$ 50 ng | 13 | $<$ 25 ng | 10 |

Number of PCR-cycles in the library preparation is dependent on gDNA input amount (in ng). Alternative 1 may be preferred when having a majority of poor-quality samples, e.g DNA from FFPE material, and alternative 2 may be preferred when having a majority of samples of high quality, e.g. cfDNA.

associated difference in performance in the library preparation and target capture process. Sample libraries with different capture panels need to go into different pools and, as a rule of thumb, sample libraries with similar library sizes should be pooled together. It is advised to pool libraries of similar size for optimal flow cell clustering effect. More than 1500 ng of total library DNA can be used for the hybridization reaction, however, using more than 4000 ng may lead to lower performance in the enrichment process.

The pool is concentrated completely until dry, using the vacuum centrifuge Concentrator Plus (Eppendorf) set on 30˚C or no heat. If several pools are handled, water is added to pools to give all pools an equal volume and to avoid pools with lower volumes to overdry during the concentration process. The hybridization mix (Twist Bioscience) is pre-heated at 65˚C for 10 min. Next, the hybridization mix is cooled to room temperature for 5 min or longer. The probe mix is prepared by 4 µl capture probes, 4 µl water or spike-in capture probes and 20 µl hybridization mix. The probe mix is heated at 95˚C for 2 min, then immediately cooled for 5 min on a cooling block, followed by equilibration to room temperature for a minimum of 5 min. The blocking mix is prepared by mixing 5 µl Blocker Solution (Twist Bioscience) and 7 µl Universal Blockers (Twist Bioscience). Next, the dried library pool is resuspended by addition of 12 µl of blocking mix, followed by mixing by pipetting 10 times (it is important to pipette along the sides of the well to make sure that all dried DNA is dissolved). The resuspended library pool is heated at 95˚C for 5 min. The probe is mixed by pipetting, and 28 µl of probe mix is added to the resuspended library pool. Mix thoroughly by pipetting since generation of bubbles should be avoided. A total of 30 µl of hybridization enhancer (Twist Bioscience) is added to the surface of the pool and incubated at 70˚C for 16–20 hours.

## Target capture, wash and amplification

A total of 100 µl of Streptavidin Binding Beads (Twist Bioscience) per capture reaction is washed three times in Binding buffer (Twist Bioscience). After the hybridization is complete, the full volume of each capture reaction including hybridization enhancer is quickly transferred to the washed Streptavidin Binding Beads and shaken (950 rpm) at room temperature for 30 min. The captured libraries are washed with 200 µl Wash Buffer 1 (Twist Bioscience) at room temperature, then three washes at 48˚C for 5 min using 48˚C pre-heated Wash Buffer 2 (Twist Bioscience) are performed. The DNA and beads are mixed with 22.5 µl of molecular biology grade water (with streptavidin binding beads remaining). Captured libraries need PCR amplification prior to sequencing. The size of a capture panel directs the number of PCR cycles needed for each pool. The streptavidin-binding bead slurry is mixed with 1x KAPA HiFi Hot-Start ReadyMix and xGen library amplification primer mix (0.5 mM, Integrated DNA Technologies) in a total volume of 50 µl. The PCR reaction is run in a thermal cycler according to the following program: initial denaturation at 98˚C for 45 sec, X cycles (defined in Table 3) of: denaturation at 98˚C for 15 sec, annealing at 60˚C for 30 sec, elongation at 72˚C for 30 sec, and a final elongation at 72˚C for 1 min. A final bead purification is then performed with 1.8X sample-to-bead ratio of AMPure XP beads (Beckman Coulter), binding at room temperature

**Table 3. Recommended number of PCR cycles after hybridization.**

| Panel size | Nr of PCR-cycles |
|---|---|
| < 0.7 Mb | 14 |
| > 0.7 Mb | 10 |
| Exome (~ 36 Mb) | 6 |

Recommended number of cycles for the post-hybridization PCR depends on the size of the panel used.

for 5 min, followed by two washes with freshly made 80% ethanol (VWR), drying 3–5 min and elution in 32 μl molecular biology grade water (Cytiva), and a final recovery of 30 μl supernatant. Quality control is performed with Quant-iT dsDNA HS assay (ThermoFisher Scientific) and TapeStation HS D1000 assay (Agilent Technologies).

## Panel descriptions

Panel 1 (GMCKsolid4.1) is a custom gene panel of 386 genes with a panel size of approximately 1.7 Mbp, synthesized by Twist Bioscience and primarily used for solid tumor analysis. The four FFPE samples and NA24143 were captured with this panel. Panel 2 (CG001) is a custom gene panel of the full coding sequence of 570 cancer-associated genes with a panel size of approximately 1.56 Mbp, synthesized by Twist Bioscience. The 570-gene list was compiled by aggregating targets from the Foundation One and MSK-IMPACT gene panels, complemented with a number of targets relevant for myeloid malignancies as well as well-established pharmacogenomic targets. Samples NA24143, HD798, HD799, HD803 and HD777 were captured using this panel.

## Sequencing and demultiplexing

Sequencing was done on NovaSeq 6000 (Illumina) using paired-end 150-bp readout, aiming at 40 million read pairs per sample. Demultiplexing was done using Illumina bcl2fastq2 Conversion Software v2.20 or using version v2.20.0.422 implemented on the DRAGEN server (Illumina).

## Data analysis

The data for samples FFPE-1, FFPE-2, FFPE-3, FFPE-4, and NA24143 used with panel GMCKsolid4.1 was downsampled to 30 million read pairs. The data for samples NA24143, HD798, HD799, HD803, and HD777 used with panel CG001 was downsampled to 40 million read pairs. All samples were analyzed using the custom-developed bioinformatic workflow BALSAMIC versions 6.0.1 and 8.13 [5] using reference genome hg19.

## Results

The design of the combination protocol presented in this study originated from efforts of trying to increase the library conversion efficiency of poor-quality FFPE samples. Initially, four poor-quality FFPE samples and a high-quality reference sample (NA24143), were used for a library conversion investigation where four different treatment conditions were tested (Fig 2). The aim was to investigate whether omission of enzymatic fragmentation and an extension of the end-repair step, in contrast to the standard KAPA HyperPlus protocol, increased the library conversion efficiency and thus increased data complexity. The KAPA HyperPrep and HyperPlus protocols were tested separately but also combined. Fig 2 shows that the combination protocol of HyperPlus and HyperPrep with 12.5-min fragmentation time gave the highest conversion rate for four out of five samples as measured by concentration after library conversion. This combination protocol specifically included an extended end-repair step as compared to the regular HyperPlus protocol, which has a very short end-repair incubation step (during ramping to 65 degrees). The yield was lower for the standard HyperPlus method both for fragmentation times 6.5 and 12.5 minutes. In addition, the HyperPlus protocol with 6.5 min fragmentation time did not produce high-quality libraries (no sharp peak). Omission of enzymatic fragmentation in the HyperPrep (including end-repair by default) protocol produced a library yield for samples FFPE-1 and FFPE-3 which was comparable to the

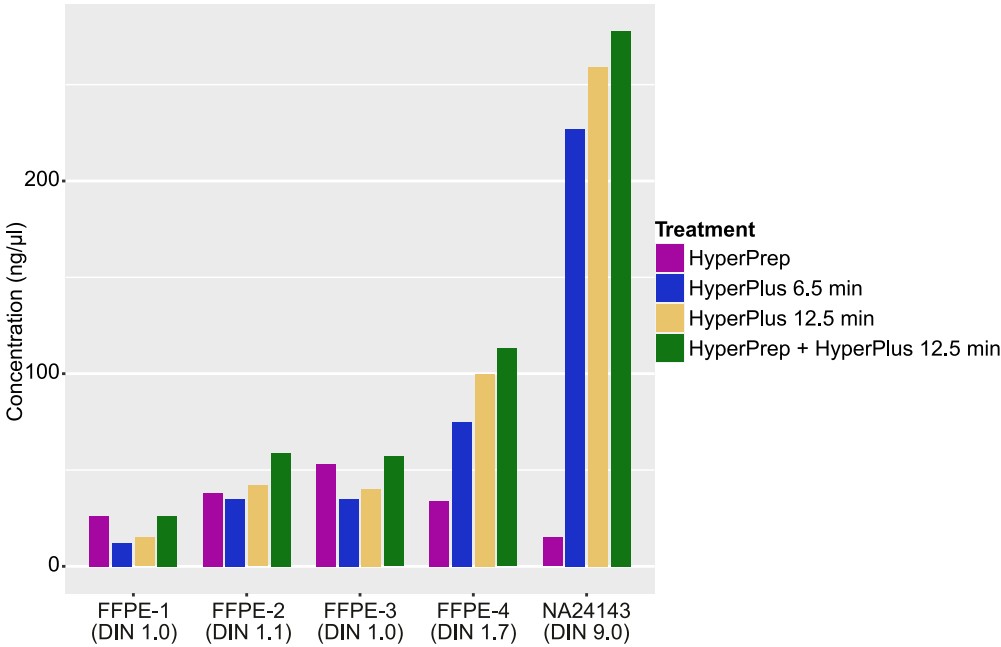

**Fig 2. Library concentrations for four library preparation strategies.** Yield in concentration (ng/µl) after library conversion for four different fragmentation and end-repair conditions; HyperPrep–no fragmentation, HyperPlus 6.5 min–fragmentation for 6.5 min, HyperPlus 12.5 min–fragmentation for 12.5 min, HyperPrep + HyperPlus–fragmentation for 12.5 min and extended end-repair. See S1 Table for data in a table format.

combination protocol, however, the method is not compatible with high quality DNA (NA24143) which requires a fragmentation step. NA24143 did not produce a library with the KAPA HyperPrep protocol.

Next, the poor FFPE samples and NA24143 were enriched with capture panel GMCKsolid4.1 and sequenced for three different treatment conditions; HyperPrep, HyperPlus and the combination of HyperPrep and HyperPlus (Fig 3). The insert size for the HyperPrep condition was longer for all FFPE samples compared to the other two treatments; HyperPlus and the combination of HyperPrep and HyperPlus (Fig 3A). However, the longer insert size for the HyperPrep protocol did not correlate with the molecular complexity of the data; the fraction duplicates (Fig 3B), median target coverage (Fig 3C) and covered bases at 150X (Fig 3D) showed an overall higher complexity for the combination protocol (HyperPrep/HyperPlus) for all FFPE samples. The data for NA24143 for HyperPlus with a short and a long end-repair was comparable, and no library could be produced for NA24143 without enzymatic fragmentation.

The combination protocol of HyperPrep and HyperPlus was performed with three different formalin-comprised reference samples (HD798, HD799, HD803), a cfDNA reference sample (HD777) as well as a high molecular weight reference sample (NA24143) run in the same library preparation setup with capture panel CG001 (Fig 4). The number of bases covered at 500X and 1000X was higher for high quality DNA (NA24143 vs HD803) as well as for higher input amounts (250 ng vs 50 ng) (Fig 4A). Percentage duplicate reads is seen to correlate with input amount as well as the quality of sample. The percentage was highest for the lowest input amount (10 ng samples) and increasing with poorer quality of sample (NA24143 vs HD803 samples) (Fig 4B). The percentage of bases covered at 500X correlates with the fraction of duplicates; the more covered bases, the less duplicates (Fig 4B and 4C). The highest median

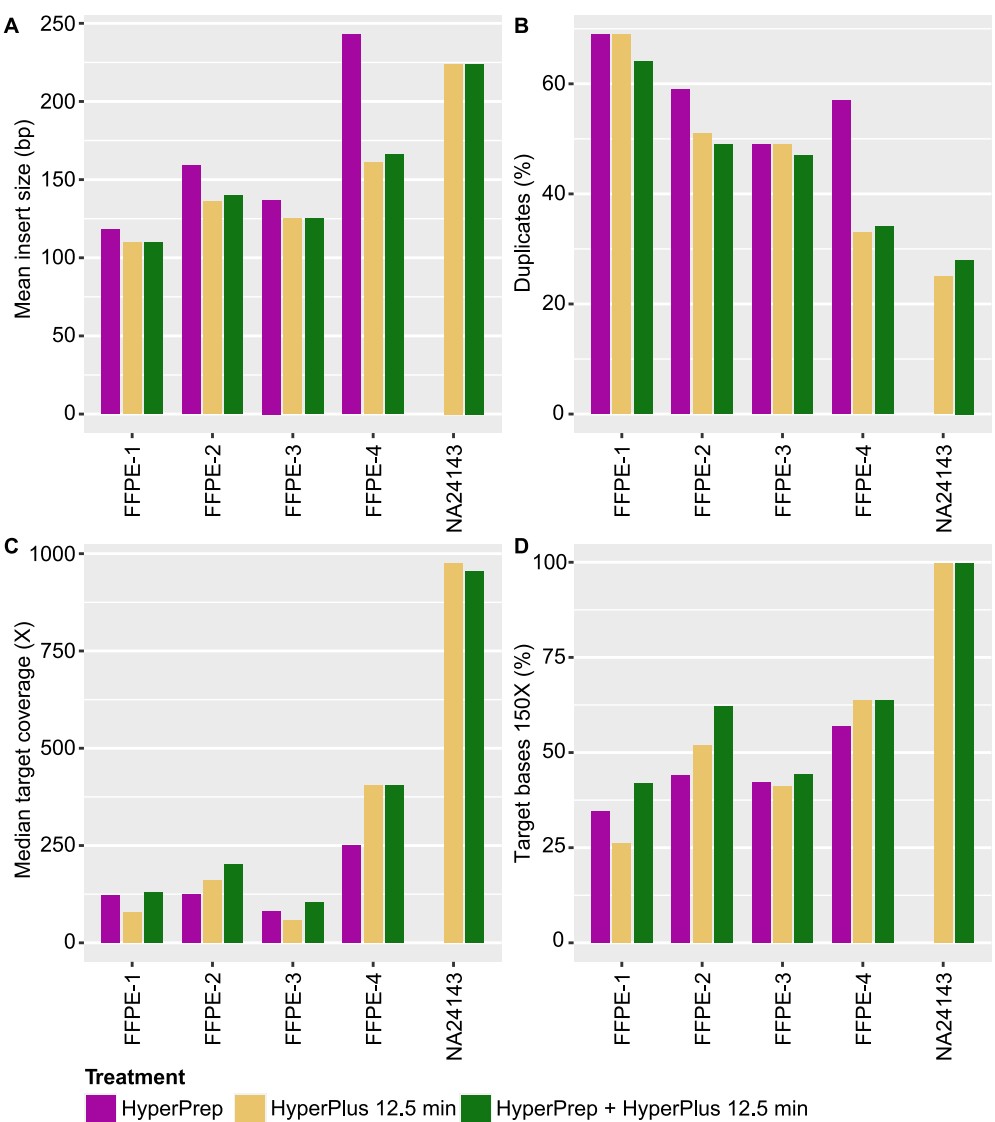

**Fig 3. Quality control data analysis for three library preparation strategies.** Analysis of quality control data for three different library preparation fragmentation and end-repair conditions; HyperPrep–no fragmentation, HyperPlus 12.5 min–fragmentation for 12.5 min, HyperPrep + HyperPlus–fragmentation for 12.5 min and extended end-repair. Mean insert size (A), duplication rate (B), median target coverage (C) and percent target bases 150X (D) in comparison to treatment conditions were investigated. Panel GMCKsolid4.1 was used. See S1 Table for data in a table format.

coverage was observed for the samples with highest quality and the highest input amount (Fig 4C). Insert size was shorter for the formalin-comprised reference samples of poorer quality (Fig 4D). The median target coverage and the percentage covered bases at 500X were lower for cfDNA 10 ng input amount compared to NA24143 10 ng input material (Fig 4A and 4C).

It was also investigated if all eleven variants previously confirmed by ddPCR could be detected in the formalin-comprised control samples (Table 4). All variants could be detected in HD798 (50 and 250 ng) and HD799 (50 ng), however the *EGFR*:p.L858R (*EGFR*, epidermal growth factor receptor), the *EGFR*:p.T790M and NRAS:p.Q61K (*NRAS*, NRAS proto-onco-gene) variants were not detected in HD799-250ng, HD803-50ng, and HD803-250ng

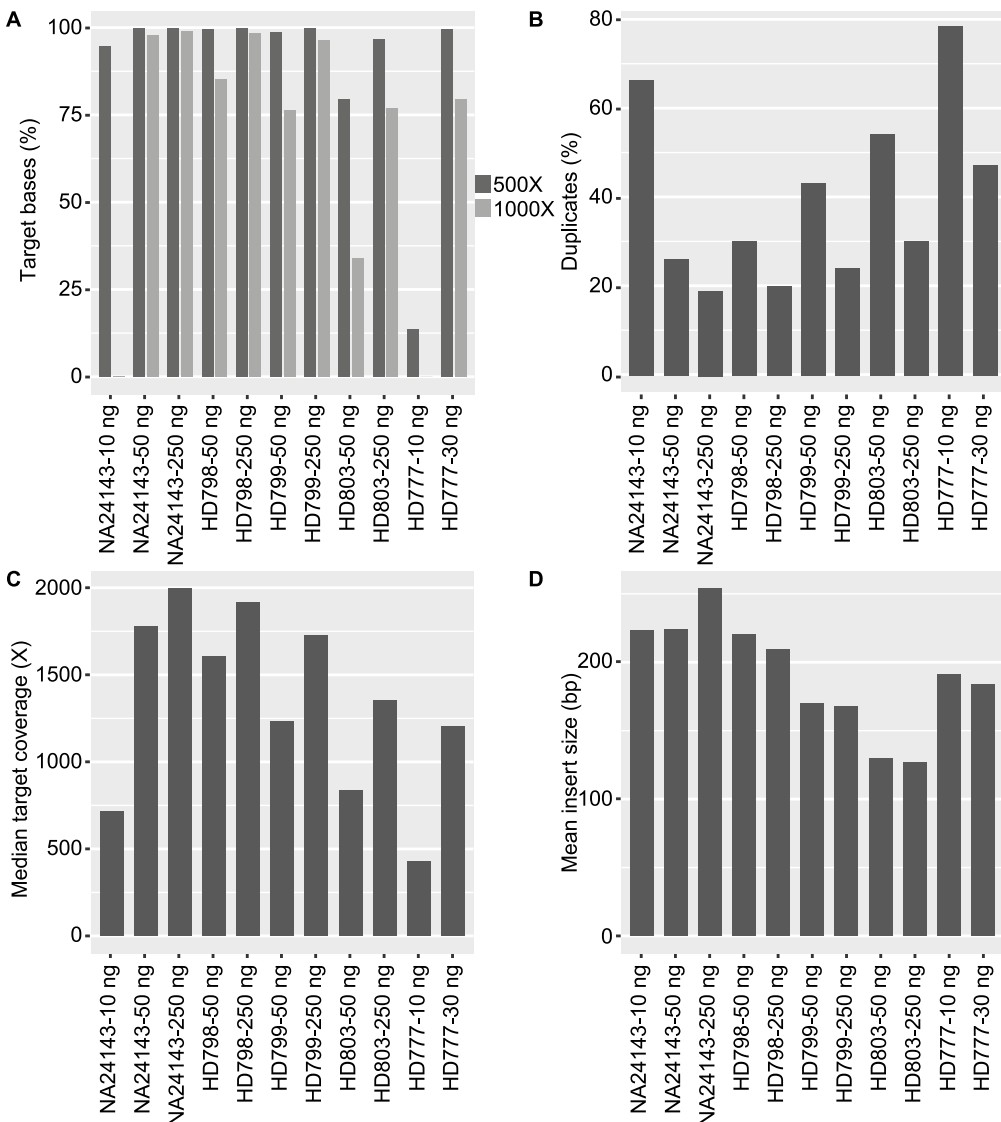

**Fig 4. Quality control data analysis for samples prepared with the combination protocol.** Five different samples (NA24143 (high molecular weight), HD798, HD799, HD803 (formalin-comprised) and HD777 (cfDNA)) at 10, 30, 50 and 250 ng input amounts were analyzed with the combination protocol (HyperPrep + HyperPlus–fragmentation for 17.5 min and extended end-repair). Data shows covered bases (percent target bases at 500 and 1000X, (A), duplication rate (B), as well as median target coverage (C) and mean insert size (D). Panel CG001 was used. See S2 Table for data in a table format.

respectively, by the bioinformatic pipeline. However, the variants could be seen in the Integrative Genomics Viewer (IGV) upon manual inspection.

It was also investigated if all eight variants previously confirmed by ddPCR could be detected in the cfDNA control sample HD777 (Table 5). All variants could be detected in HD777.

## Discussion

We have described a protocol that can handle different sample types as well as different sample qualities and quantities for parallel library preparation and target enrichment using Illumina

**Table 4. Variants detected in the formalin-comprised control samples HD798, HD799 and HD803 prepared with the combination protocol.**

| Gene:p.aa | AF Expected | HD798-50ng AF Vardict | HD798-250ng AF Vardict | HD799-50ng AF Vardict | HD799-250ng AF Vardict | HD803-50ng AF Vardict | HD803-250ng AF Vardict |
|---|---|---|---|---|---|---|---|
| *BRAF*:p.V600E | 0.105 | 0.120 | 0.111 | 0.117 | 0.117 | 0.120 | 0.118 |
| *KIT*:p.D816V | 0.1 | 0.157 | 0.080 | 0.088 | 0.093 | 0.090 | 0.107 |
| *EGFR*:p.E746_A750 delELREA | 0.02 | 0.012 | 0.009 | 0.012 | 0.018 | 0.026 | 0.018 |
| *EGFR*:p.L858R | 0.03 | 0.038 | 0.039 | 0.043 | 0.036[a] | 0.042 | 0.040 |
| *EGFR*:p.T790M | 0.01 | 0.012 | 0.010 | 0.010 | 0.009 | 0.011[b] | 0.009 |
| *EGFR*:p.G719S | 0.245 | 0.233 | 0.235 | 0.212 | 0.231 | 0.228 | 0.204 |
| *KRAS*:p.G13D | 0.15 | 0.145 | 0.147 | 0.146 | 0.126 | 0.157 | 0.137 |
| *KRAS*:p.G12D | 0.06 | 0.079 | 0.056 | 0.063 | 0.052 | 0.055 | 0.069 |
| *NRAS*:p.Q61K | 0.125 | 0.144 | 0.122 | 0.119 | 0.110 | 0.114 | 0.120[c] |
| *PIK3CA*:p.E545K | 0.09 | 0.153 | 0.093 | 0.058 | 0.083 | 0.081 | 0.071 |
| *PIK3CA*:p.H1047R | 0.175 | 0.170 | 0.179 | 0.186 | 0.170 | 0.207 | 0.195 |
| Number of variants | | 3266 | 3048 | 3147 | 2959 | 2323 | 2542 |

Variant allele frequencies detected in HD798, HD799 and HD803 samples (50 ng and 250 ng input amount) for BRAF, B-Raf proto-oncogene, serine/threonine kinase; KIT, KIT proto-oncogene, receptor tyrosine kinase; EGFR, epidermal growth factor receptor; KRAS, KIT proto-oncogene, receptor tyrosine kinase; NRAS, NRAS proto-oncogene, GTPase; PIK3CA, phosphatidylinositol-4,5-bisphosphate 3-kinase catalytic subunit alpha and the total number of variants. See S3 Table the Quality score from Vardict.

[a]Data from IGV (152 out of 4206 counts, 3.6%).
[b]Data from IGV (31 out of 2930 counts, 1%).
[c]Data from IGV (251 out of 2084 counts, 12%).

short read sequencing. The concept of the combination protocol originated from attempts to improve the library conversion rate for poor quality FFPE samples which gave rise to a protocol that not only improved the conversion efficiency for FFPE samples but also made it possible to include different kinds of samples in the same workflow. This enables high throughput and short turnaround times since many different sample types can be handled simultaneously. This type of highly effective production workflow is advantageous in for example clinical diagnostic settings.

**Table 5. Variants detected in the cfDNA control sample HD777 prepared with the combination protocol.**

| Gene:p.aa | AF Expected | HD777-10ng AF Vardict | HD777-30ng AF Vardict |
|---|---|---|---|
| *EGFR*:p.L858R | 0.050 | 0.033 | 0.046 |
| *EGFR*:p.E746_A750 delELREA | 0.050 | 0.057 | 0.053 |
| *EGFR*:p.T790M | 0.050 | 0.039 | 0.045 |
| *EGFR*:A767_V769dup | 0.050 | 0.050 | 0.047 |
| *KRAS*:p.G12D | 0.050 | 0.084 | 0.056 |
| *NRAS*:p.Q61K | 0.050 | 0.088 | 0.069 |
| *NRAS*:p.A59T | 0.050 | 0.073 | 0.068 |
| *PIK3CA*:p.E545K | 0.050 | 0.080 | 0.061 |
| Number of variants | | 2127 | 2452 |

Variant allele frequencies detected in the HD777 sample (10 ng and 30 ng input amounts) for *EGFR*, epidermal growth factor receptor; *KRAS*, KIT proto-oncogene, receptor tyrosine kinase; *NRAS*, NRAS proto-oncogene, GTPase; *PIK3CA*, phosphatidylinositol-4,5-bisphosphate 3-kinase catalytic subunit alpha and the total number of variants. See S4 Table the Quality score from Vardict.

The extension of the end-repair step to the standard KAPA HyperPlus protocol increased the library conversion rate for FFPE samples and had no negative impact on high quality DNA. We hypothesize that some damaged DNA molecules in an FFPE sample cannot efficiently be converted to library molecules by enzymatic fragmentation, instead, they can be rescued and turned into full library molecules that can be sequenced by extending the end-repair step (Fig 2). This step is by default not included in the KAPA HyperPlus protocol but can be added by an extra incubation at 65 degrees Celsius during 30 minutes, no new reagents need to be added. The fragmentation step reduces the insert size for FFPE samples in varying degrees, however, the complexity of the data is still better for FFPE samples that have been fragmented (Fig 3). This is probably explained by more but slightly shorter fragments that can be converted into library molecules by including both fragmentation and a longer end-repair process. The extension of the end-repair steps makes it also possible to include cfDNA samples which should not be fragmented due to the inherently short molecular length (Fig 4). The end-repair step is essential for cfDNA to produce blunt ends prior to the a-tailing and adaptor ligation steps.

Recall of variants for the formalin-comprised samples as well as the cfDNA control samples was investigated. All variants could be detected except for the *EGFR*: *p.T790M*, *EGFR*: *p.L858R* and *NRAS*: *p.Q61K* mutations in HD799-250ng, HD803-50ng, and HD803-250ng, however these variants could still be viewed in IGV. The *NRAS*: *p.Q61K* variant missed by Vardict could be found with DNAscope. All variants could be detected in the HD777 sample. Low level variants typically require the use of UMI sequences for noise reduction which are included in this protocol for all libraries.

The protocol involves optional settings such as fragmentation time depending on the desired insert size best suited for the application and the sequencing read length used. The fragmentation time should not be too short, since short fragmentation time reduces the conversion rate for both high quality samples as well as poor quality FFPE samples (Fig 2). The fragmentation efficiency should preferably be monitored over time since the activity of the fragmentation enzyme can vary between batches of the library preparation kit. Depending on the major sample types and DNA quantities intended to be used or available, the number of PCR cycles and adapter amount can be adjusted. Reducing the number of PCR cycles can lead to an inability to capture the libraries of poor FFPE samples, but too many PCR cycles can result in a higher percentage of duplicates. The ligation time is relatively short, so in order to keep conversion rates high, the adapter content (defined as the adapter-to-insert ratio) should be high. If adapter dimers are problematic, the adapter amount can be reduced but it should be kept in mind that the ligation time then may need to be extended.

Different samples may need to be sequenced to different depths depending on the level of allele frequencies intended to be investigated. The sequencing depth can be regulated during different stages of the target enrichment and sequencing process. First, individual libraries with the same capture panel can be pooled together in different ratios against each other in a capture pool. This will result in more reads for specific samples having a larger proportion of the total capture library pool. Second, different target capture pools can also be pooled in different ratios against each other during sequencing preparation depending on the read depth desired. This will result in more reads for capture pools with larger proportion of the final sequencing pool. Libraries generated from FFPE material may result in lower number of reads than desired, hence it can be advantageous to pool only four individual libraries in such capture reactions, avoiding high costs associated with re-sequencing of the entire pool if a certain performance level is required.

We have presented a flexible and cost-effective library preparation and target enrichment method for all relevant clinical sample types, including challenging sample categories such as

FFPE and cfDNA samples, input quantities and capture panel designs (target sizes). Different sample libraries can be pooled with the same capture panel or into the same sequencing pool, and the sequencing depth can be modulated during pooling to capture and/or sequencing, which enables an efficient workflow and variant calling at different sensitivities. The method can be performed manually but can also be automated on a liquid handling system, and the protocol presented here has been set up on a Hamilton NGS Star system. This action is a direct response to an expected strong increase in sample categories and numbers as NGS moves into routine diagnostic settings at a high scale.

## Supporting information

**S1 Table. Data for five samples prepared with four different library preparation strategies.** The data includes DIN value, concentration, mean insert size, fraction duplicates, median target coverage and percent (pct) target bases covered at 150X for samples (FFPE-1, FFPE-2, FFPE-3, FFPE-4 and NA24143) prepared with library preparation strategies; HyperPrep (no fragmentation), HyperPlus fragmentation 6.5 min, HyperPlus fragmentation 12.5 min, combination of HyperPrep and Hyper Plus fragmentation 12.5 min.
(XLSX)

**S2 Table. Data for five samples prepared with the combination protocol.** The data includes pipeline version, mean insert size, fraction duplicates, median target coverage, and percent (pct) target bases at 500 and 1000X, for samples NA24143, HD798, HD799, HD803 and HD777 at 10, 30, 50 and 250 ng input amounts with the combination protocol (HyperPrep + HyperPlus).
(XLSX)

**S3 Table. Variants detected in the formalin-comprised control samples prepared with the combination protocol.** The data includes Gene:p.aa, AF Expected, AF Vardict and Quality score from Vardict for HD798, HD799, HD803 at 50 and 250 ng input amounts with the combination protocol (HyperPrep + HyperPlus).
(XLSX)

**S4 Table. Variants detected in the cfDNA control sample prepared with the combination protocol.** The data includes Gene:p.aa, AF Expected, AF Vardict and Quality score from Vardict for HD777 at 10 and 30 ng input amounts with the combination protocol (HyperPrep + HyperPlus).
(XLSX)

## Acknowledgments

We acknowledge Johan Lindberg (Karolinska Institutet) for the design of the GMCKsolid panel.

## Author Contributions

**Conceptualization:** Anna Lyander, Anna Gellerbring, Valtteri Wirta.

**Data curation:** Anna Lyander, Anna Gellerbring, Moa Hägglund, Keyvan Elhami.

**Formal analysis:** Anna Lyander, Anna Gellerbring, Moa Hägglund, Keyvan Elhami.

**Investigation:** Anna Lyander, Anna Gellerbring.

**Methodology:** Anna Lyander, Anna Gellerbring, Valtteri Wirta.

**Visualization:** Anna Lyander, Anna Gellerbring, Moa Hägglund.

**Writing – original draft:** Anna Lyander, Anna Gellerbring.

**Writing – review & editing:** Anna Lyander, Anna Gellerbring, Moa Hägglund, Keyvan Elhami, Valtteri Wirta.

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
