## [Decision Letter · Decision Letter 0]

2 Jan 2024

PONE-D-23-41807NGS method for parallel processing of high quality, damaged or fragmented input material using target enrichmentPLOS ONE

Dear Dr. Lyander,

Thank you for submitting your manuscript to PLOS ONE. After careful consideration, we feel that it has merit but does not fully meet PLOS ONE’s publication criteria as it currently stands. Therefore, we invite you to submit a revised version of the manuscript that addresses the points raised during the review process.

We look forward to receiving your revised manuscript.

Kind regards,

Elingarami Sauli, PhD

Academic Editor

PLOS ONE

Journal Requirements:

   "VW has received reimbursement of travel costs and speaker’s honoraria from Illumina."

3. In the online submission form, you indicated that "All relevant data are contained within the manuscript. Data of Figures 2-4 can be provided upon request."

Additional Editor Comments:

When responding to reviewer comments, please make sure to clearly elaborate/explain differences in results from FFPE1-4 and Horizon reference samples. Also, differentiate your research article from a lab protocol, based on your provided data.

Reviewers' comments:

Reviewer's Responses to Questions

**Comments to the Author**

1. Is the manuscript technically sound, and do the data support the conclusions?

Reviewer #1: Partly

Reviewer #2: Yes

Reviewer #3: Yes

2. Has the statistical analysis been performed appropriately and rigorously? 

Reviewer #1: N/A

Reviewer #2: Yes

Reviewer #3: Yes

3. Have the authors made all data underlying the findings in their manuscript fully available?

Reviewer #1: Yes

Reviewer #2: Yes

Reviewer #3: Yes

4. Is the manuscript presented in an intelligible fashion and written in standard English?

Reviewer #1: Yes

Reviewer #2: Yes

Reviewer #3: Yes

5. Review Comments to the Author

Reviewer #1: Lyander et al. “NGS method for parallel processing of high quality, damaged or fragmented input

material using target enrichment”

In this article, the authors describe a procedure for NGS library preparation form different types of samples.

I have the following questions/comments:

1. Were the same input amounts used for FFPE1-4 and the Horizon reference samples?

2. What QC metrics are used for the 2 panels described by the authors? The reported duplication rates, target coverage and %target bases for the FFPE samples would, in many clinical laboratories, be considered as failures or not passing QC.

3. Can the authors comment on the differences in QC metrics in Figure 3 and 4, which are using different samples and panels. The results seem better with the Horizon reference samples and panel 2. Do the authors think this is due to the difference in panels/probes, or due to a difference between “real” samples and commercial reference samples?

4. Table 4. Can the authors also provide the quality score for the variant calls?

5. Table 4. Can the authors include data for HD777 in the table.

6. Can the authors also include the total number of variants detected for the samples? In many cases, detecting the expected variants is not the problem with compromised samples, but instead with high duplication rates, early PCR errors result in a high number of artifactual calls and a large number of false positive variants.

Reviewer #2: The manuscript entitled "NGS method for parallel processing of high quality, damaged or fragmented input

material using target enrichment" described a joint procedure for preparing enriched

DNA libraries from high molecular weight DNA and DNA from formalin-fixed, paraffinembedded tissue, fresh frozen tissue material, as well as cell-free DNA.

- The Authors should provide the expand forms for all acronyms, including gene acronyms, through the text when they first appear.

- Gene acronyms should be written in italics.

Reviewer #3: The sutdy was conducted in a very relevant topic. The author very elaborately detailed an alterantive in a challenging issuee. I onlye suggest if author provide a pictorial diagram outlining the steps for the workflow, highlighting important QC steps which one should be aware of, will be helpful for the readers.

6. PLOS authors have the option to publish the peer review history of their article (what does this mean?). If published, this will include your full peer review and any attached files.

Reviewer #1: No

Reviewer #2: No

Reviewer #3: **Yes: **Subit Barua

---

## [Author Response · Author response to Decision Letter 0]

15 Feb 2024

Response to reviewers

Author reply: The manuscript has been edited according to the style requirements.

 "VW has received reimbursement of travel costs and speaker’s honoraria from Illumina."

Author reply: The Competing Interests section has been clarified and added to the cover letter.

3. In the online submission form, you indicated that "All relevant data are contained within the manuscript. Data of Figures 2-4 can be provided upon request."

Author reply: The data has been added as supplementary data; S1 Table (Fig 2-3) and S2 Table (Fig 4).

Author reply: The ethics statement has been moved to the Materials and Methods section.

Author reply: The references have been modified according to the Vancouver style.

Additional Editor Comments:

When responding to reviewer comments, please make sure to clearly elaborate/explain differences in results from FFPE1-4 and Horizon reference samples. Also, differentiate your research article from a lab protocol, based on your provided data.

Author reply: This has been done accordingly.

Comments to the Author

5. Review Comments to the Author

Reviewer #1: Lyander et al. “NGS method for parallel processing of high quality, damaged or fragmented input

material using target enrichment”

In this article, the authors describe a procedure for NGS library preparation form different types of samples.

I have the following questions/comments:

1. Were the same input amounts used for FFPE1-4 and the Horizon reference samples?

Author reply: The input amounts used for the samples have been added to the Methods section, in “Enzymatic fragmentation of gDNA”.

2. What QC metrics are used for the 2 panels described by the authors? The reported duplication rates, target coverage and %target bases for the FFPE samples would, in many clinical laboratories, be considered as failures or not passing QC.

Author reply: This could be considered failed based on the clinical application used. To improve the QC metrics a larger amount of DNA from the FFPE material could be used. In addition, deeper sequencing can improve the QC metrics. The samples used in this case were poor quality DNA intended to show how data could be improved with an alternative laboratory protocol, and may not be fully representative for all clinical cases.

3. Can the authors comment on the differences in QC metrics in Figure 3 and 4, which are using different samples and panels. The results seem better with the Horizon reference samples and panel 2. Do the authors think this is due to the difference in panels/probes, or due to a difference between “real” samples and commercial reference samples?

Author reply: The results seem better with the Horizon reference samples compared to the FFPE samples. This is likely due to the difference in quality between the commercial formalin-comprised Horizon samples (not FFPE material) HD798, HD799 and HD803 with a DIN value of 7.3, 3.9 and 1.7, respectively, and the formalin fixed paraffin embedded patient samples with DIN 1.0, 1.1, 1.0, 1.7.

4. Table 4. Can the authors also provide the quality score for the variant calls?

Author reply: The quality scores for the variant calls have been added to supplementary tables 3 and 4.

5. Table 4. Can the authors include data for HD777 in the table.

Author reply: This has been added in table 5. 

6. Can the authors also include the total number of variants detected for the samples? In many cases, detecting the expected variants is not the problem with compromised samples, but instead with high duplication rates, early PCR errors result in a high number of artifactual calls and a large number of false positive variants.

Author reply: The total numbers of variants detected have been added to supplementary tables 3 and 4.

Reviewer #2: The manuscript entitled "NGS method for parallel processing of high quality, damaged or fragmented input

material using target enrichment" described a joint procedure for preparing enriched

DNA libraries from high molecular weight DNA and DNA from formalin-fixed, paraffinembedded tissue, fresh frozen tissue material, as well as cell-free DNA.

- The Authors should provide the expand forms for all acronyms, including gene acronyms, through the text when they first appear.

- Gene acronyms should be written in italics.

Author reply: This has been changed accordingly. 

Reviewer #3: The sutdy was conducted in a very relevant topic. The author very elaborately detailed an alterantive in a challenging issuee. I onlye suggest if author provide a pictorial diagram outlining the steps for the workflow, highlighting important QC steps which one should be aware of, will be helpful for the readers.

Author reply: Criteria of ≥ 187.5 ng (under Quant-iT) has been added to figure 1.

---

## [Decision Letter · Decision Letter 1]

13 May 2024

NGS method for parallel processing of high quality, damaged or fragmented input material using target enrichment

PONE-D-23-41807R1

Dear Dr. Anna,

We’re pleased to inform you that your manuscript has been judged scientifically suitable for publication and will be formally accepted for publication once it meets all outstanding technical requirements.

Kind regards,

Elingarami Sauli, PhD

Academic Editor

PLOS ONE

Additional Editor Comments:

The decision to accept this submission was reached based on  the fact that, the authors  have responded on a previous query regarding the input amounts for FFPE1-4 and the Horizon reference used for the samples, which you have added to the Methods section, in “Enzymatic fragmentation of gDNA”.

As regards to raised query on QC metrics used for the 2 panels, as described on duplication rates, target coverage and %target bases for the FFPE samples, which would in many clinical laboratories be considered as failures or not passing QC, the authors have agreed with this and indicated that, in order to improve the QC metrics, a larger amount of DNA from the FFPE material could be used. In addition, the authors affirm that, deeper sequencing can improve the QC metrics. The authors have further clarified that, samples used in this case were of poor quality DNA intended to show how data could be improved with an alternative laboratory protocol, and may not be fully representative for all clinical cases.

As regards to a query on differences in QC metrics in Figure 3 and 4, which are using different samples and panels. The results seem better with the Horizon reference samples and panel 2. The authors have agreed that, yes the results seem better with the Horizon reference samples compared to the FFPE samples, which is possibly/likely due to the difference in quality between the commercial formalin-comprised Horizon samples (not FFPE material) HD798, HD799 and HD803 with a DIN value of 7.3, 3.9 and 1.7, respectively, and the formalin fixed paraffin embedded patient samples with DIN 1.0, 1.1, 1.0, 1.7.

Reviewers' comments:

Reviewer's Responses to Questions

**Comments to the Author**

1. If the authors have adequately addressed your comments raised in a previous round of review and you feel that this manuscript is now acceptable for publication, you may indicate that here to bypass the “Comments to the Author” section, enter your conflict of interest statement in the “Confidential to Editor” section, and submit your "Accept" recommendation.

Reviewer #2: All comments have been addressed

Reviewer #4: (No Response)

2. Is the manuscript technically sound, and do the data support the conclusions?

Reviewer #2: Yes

Reviewer #4: Partly

3. Has the statistical analysis been performed appropriately and rigorously? 

Reviewer #2: Yes

Reviewer #4: N/A

4. Have the authors made all data underlying the findings in their manuscript fully available?

Reviewer #2: Yes

Reviewer #4: Yes

5. Is the manuscript presented in an intelligible fashion and written in standard English?

Reviewer #2: Yes

Reviewer #4: Yes

6. Review Comments to the Author

Reviewer #2: The Authors have addressed all my concerns and I have no further comments.

The manuscript is suitable for publication.

Reviewer #4: Developing a protocol that is optimized for all sample types is relevant to clinical practice. Therefore, testing real world clinical samples (as many as possible) to demonstrate that the protocol is robust with accuracy, reproducibility and limits of detections acceptable for clinical practice is desired.

1. It remains not entirely clear to me if cf DNA has been subjected to enzymatic fragmentation in the combination protocol. In Figure 1, it states that “the first step is enzymatic fragmentation, which is performed for all samples except cfDNA samples, and the following end-repair/A-tailing step is a combination of KAPA HyperPrep and HyperPlus protocols (hereafter denoted combination protocol)”. However, in page 7, line 149, under the section of enzymatic fragments of gDNA, HD777, a cf DNA was included in the experiment. Confusedly, in page 5 under the section of Library preparation, it states that “for a combined preparation of both gDNA and cfDNA, the Hyperplus kit (includes enzymatic fragmentation) is used. However, if there solely cfDNA samples, the HyperPre kit (without enzymatic fragmentation) should be used instead”. Please make consistent statements throughout the manuscript and clearly indicate if one or two kits are required for the proposed combination protocol.

2. Table 2. Could the authors be more specific about the selection of alternative 1 or alternative 2? The selection based on “majority” of samples is vague and imprecise. Is 5 out of 8 or 7 out of 8 considered “majority”?

3. My main concern is that the combination protocol may introduce false positive variant/artificial calls or false negative results in the FFPE samples due to over-treatment. For example in Table 4, NRAS: Q61K with VAF of 12% was not detected in one of the samples but required manual inspection of IGV to find the variant. I suggest the authors to evaluate more clinical FFPE samples with known variants detected by orthogonal methods. Presenting the performance data of applying combination protocol on FFPE samples including sensitivity, specificity, reproducibility and percent of failure rate could significantly strengthen the conclusions of this manuscript.

4. Using cfDNA sample with 5% VAF is not ideal to evaluate the performance of a protocol designing for clinical practice. Clinically, the lower limit of detection of cfDNA is in general set at 0.1% VAF. Including cfDNA samples with lower VAFs for evaluation is highly recommended.

7. PLOS authors have the option to publish the peer review history of their article (what does this mean?). If published, this will include your full peer review and any attached files.

Reviewer #2: No

Reviewer #4: No

---

## [Editor Report · Acceptance letter]

17 May 2024

PONE-D-23-41807R1 

PLOS ONE

Dear Dr. Lyander, 

I'm pleased to inform you that your manuscript has been deemed suitable for publication in PLOS ONE. Congratulations! Your manuscript is now being handed over to our production team.

Kind regards, 

on behalf of

Dr. Elingarami Sauli 

Academic Editor

PLOS ONE